# "Urethral-Sparing" Robotic Radical Prostatectomy: Critical Appraisal of the Safety of the Technique Based on the Histologic Characteristics of the Prostatic Urethra

**Anastasios D. Asimakopoulos** [1,2,*], **Filippo Annino** [2], **Gaia Colalillo** [1], **Richard Gaston** [3], **Thierry Piechaud** [3], **Alessandro Mauriello** [4], **Umberto Anceschi** [5] and **Filippo Borri** [6]

1   Urology Unit, Fondazione PTV Policlinico Tor Vergata, 00133 Rome, Italy
2   Urology Unit, Azienda USL Toscana Sud-Est, San Donato Hospital, 52100 Arezzo, Italy
3   Unit of Urology, Clinique Saint-Augustin, 33074 Bordeaux, France
4   Pathology, Department of Experimental Medicine and Surgery, University of Rome Tor Vergata, 00133 Rome, Italy
5   Department of Urology, IRCCS Regina Elena National Cancer Institute, 00144 Rome, Italy
6   Anatomic Pathology, Azienda USL Toscana Sud-Est, San Donato Hospital, 52100 Arezzo, Italy
*   Correspondence: tasospao2003@yahoo.com; Tel.: +39-06-2090-2974; Fax: +39-06-2090-2975

**Abstract:** Background: The prostatic urethra (PU) is conventionally resected during robot-assisted radical prostatectomy (RALP). Recent studies demonstrated the feasibility of the extended PU preservation (EPUP). Aims: To describe the histologic features of the PU. Methods: The PU was evaluated using cystoprostatectomy and RALP specimens. Cases of PU infiltration by prostate cancer or distortion by benign hyperplastic nodules were excluded. The thickness of the chorion and distance between the urothelium and prostate glands were measured. Prostate-specific antigen expression in the PU epithelium was evaluated with immunohistochemistry. Descriptive statistics were used. Results: Six specimens of PU were examined. Histologically, the following layers of the PU were observed: (1) urothelium with basal membrane, (2) chorion, and (3) prostatic peri-urethral fibromuscular tissue. The chorion measures between 0.2 and 0.4 mm. There is not a distinct urethral muscle layer, but rather muscular fibers that originate near the prostatic stroma and are distributed around the PU. This muscular tissue appears to be mainly represented in the basal and apical urethra, but not in the middle urethra. The mean distance between the chorion and prostatic glands is 1.74 mm, with significant differences between base of the prostate, middle urethral portion, and apex (2.5 vs 1.49 vs 1.23 mm, respectively). PSA-expressing cells are abundant in the PU epithelium, coexisting with urothelial cells. Conclusions: The exiguity of thickness of the PU chorion, short distance from glandular tissue, and coexistence of PSA-expressing cells in the epithelium raise important concerns about the oncologic safety of EPUP.

**Keywords:** prostate cancer; incontinence; prostatectomy; urethra; urothelium

## 1. Introduction

Radical prostatectomy (RP) is a common modality for the treatment of localized PCa [1]. Even in experienced hands, incontinence is one of the most feared complications of RP [2], because it may strongly affect the quality of life of patients [3].

While several studies have evaluated the effect of bladder neck preservation in early return of urinary continence [4], little is known about the technical feasibility and functional outcomes of prostatic urethral preservation (PUP). Some benefit in early recovery of continence has been suggested by retrospective series in which a portion of the apical urethra is spared during open RP [5]. Recently, Nunez Bragayrac LA et al. [6] reported the outcomes of 48 consecutive, single surgeon robot-assisted RP (RALP) operations using extended prostatic urethra preservation (EPUP). Urethral dissections typically spanned the

prostatic apex to mid-gland or base. Their approach "telescoped" the urethra muscle and continued until:

(1) the plane failed to develop easily,
(2) the urethra muscle became too thin (reaching mucosa or tearing),
(3) or the dissection reached the transected bladder neck (i.e., complete urethral sparing).

The authors concluded that EPUP is technically feasible during RARP and is associated with faster continence recovery with respect to standard RALP, in which the PU is conventionally resected [6].

However, in the same study, only two patients had preservation of the entire PU, raising concerns about the reproducibility of the technique, even in the hands of the same surgeon. At the completion of the vesicourethral anastomosis, the surgeon added some stitches in order to shorten the preserved urethra and avoid tearing; thus, no knowledge was provided either on eventual kinking of the preserved urethral stump or on the vascularization of the preserved segment. Finally, nearly 8% of EPUP patients had a detectable PSA (>0.1 ng/dL) at a median follow-up of 12 months.

The aim of our study was to provide a histologic description of the PU in order to evaluate the safety of its complete preservation.

## 2. Materials and Methods

Twenty-four prostatic specimens of RP and cystoprostatectomy, performed for the treatment of prostate and muscle-invasive bladder cancer, respectively, were obtained from routine urological clinical practice and were histologically analyzed. Eighteen of them were characterized by PU distortion by hyperplastic nodules, or PU neoplastic involvement, and were excluded from the final analysis. Consequently, six specimens with a "reliable" depiction of the histologic characteristics of the "standard" PU were included.

All specimens were totally submitted and sampled according to international sampling protocols (Table 1) [7]. Prostate weight and the three dimensions of the gland were measured, and specimens were fixed by immersion in 10% formalin for at least 24–36 h.

**Table 1.** Pathologic evaluation of the specimens.

| | |
|---|---|
| Fixation | En bloc, 10% neutral buffered formalin |
| Coating (to delineate surgical margins) | India ink. |
| Specimen processing | Specimens were step-sectioned transversely at 3–4-mm intervals. |
| Apex and base | An apical and basal shaved section, 3–4-mm thick, was truncated perpendicular to the prostatic urethra and subsequently sectioned as slices parallel to the prostatic urethra. |
| Bladder neck section | Either by sampling portions of tissue at the junction of the prostatic capsule and bladder neck, or by sampling the most proximal portion of the submitted specimen corresponding to the anatomical bladder neck. |
| Staining | Hematoxylin and eosin. |
| Intraprostatic urethra evaluation | Three axial sections of proximal (post-basal), intermediate (middle), and distal (pre-apical) part; each section further divided in four subsections, left–right and anterior–posterior. |
| Evaluated parameters | • Prostate weight, intraprostatic urethral length.<br>• Chorion thickness.<br>• Distance between epithelial basal membrane of PU and true prostatic glands. |
| Immunohistochemistry | Anti-GATA 3.<br>Anti-PSA. |

For every specimen, proximal (basal) and distal (apical) margins were sampled by sagittal sections at 3 mm intervals perpendicular to the cutting fissure; the remaining

prostate was transversally sectioned, obtaining 3–4 mm thick slices. Every prostatic slice was further divided in four parts in order to distinguish between left–right and anterior–posterior portions (Figure 1).

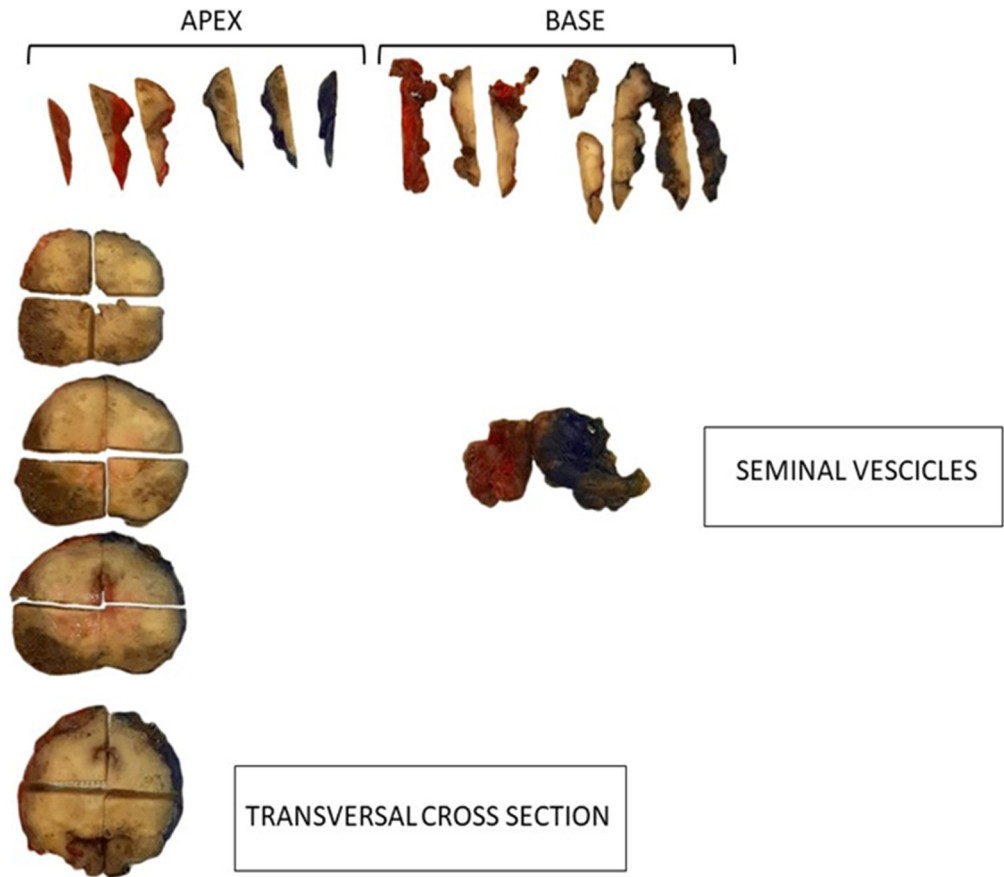

**Figure 1.** Prostatic specimen sectioned according to international grossing protocols for pathological examination.

Both the apical and basal sections were only morphologically evaluated; no measurements were performed here, due to the presence of surgical resection artefacts.

Thus, from an anatomic point of view and for practical purposes, the PU sections were divided in three zones (Figure 2):

(1) Post-basal zone: between first post-basal section and verumontanum,
(2) Middle zone: between second half of verumontanum and pre-apical sections,
(3) Pre-apical zone: remaining pre-apical sections.

Since the PU is usually characterized by irregular shape, its histological evaluation was performed considering the urethral structure in its entirety; thus, pseudostratified urothelium and umbrella cells, secretory periurethral glands sprouting in the main urethral lumen, and minor anatomical variations, such as von Brunn nests and papillary/polypoid projections, were included as main components of the urethra.

We performed a microscopic evaluation of hematoxylin- and eosin-stained histological sections, with an optical microscope Zeiss Axio Scope A1 (Carl Zeiss Microscopy GmbH, Munich (Germany)) at 10× objective lens magnification (2 mm field diameter), employing an eyepiece grid net micrometer calibration to obtain precise measurement of the thickness of the chorion, as well as the distance between the PU urothelial basal membrane and true prostatic glands. A morphological evaluation of peri-urethral muscular layers across the entire urethral sections was also performed. All the measures obtained were expressed in millimeters.

Finally, an immunohistochemical evaluation of the PU epithelium was performed with antibodies anti-GATA 3 (mAb clone L50-823, Ventana Medical Systems, Tucson, Arizona and anti-PSA (mAb clone ER-PR8 760-4271, Ventana Medical Systems, Tucson, Arizona), and their expression was represented as percentage of positive epithelial cells over the total epithelial cell number in representative sections.

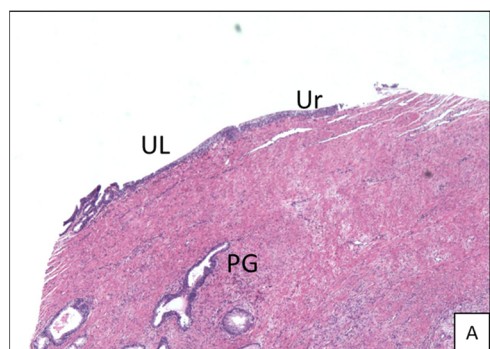
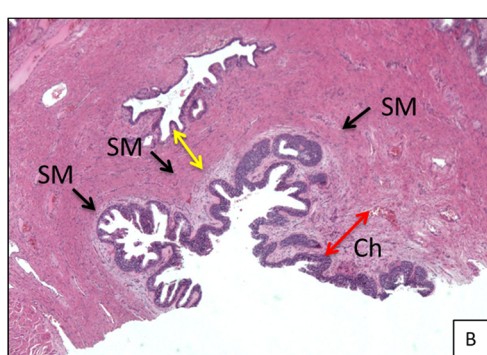
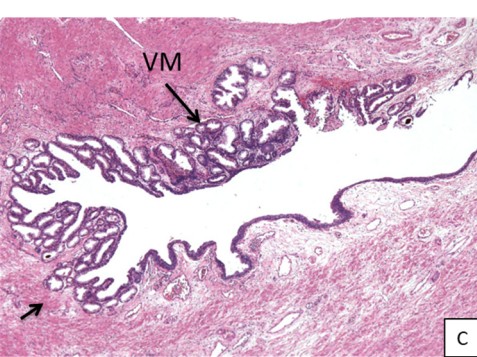

**Figure 2.** (**A**) Representative section of prebasal portion of prostatic urethra (H&E, 4× magnification). (**B**) Representative section of prebasal portion of prostatic urethra: fibromuscular tissue is well represented and fibers are disposed in a sleeve around the urethra (arrow), as a continuation of detrusor muscle (H&E, 4× magnification); red double arrow indicates chorion; yellow double arrow indicates distance between chorion and prostatic glands. (**C**) Middle portion of urethra: in this picture part of the verumontanum is represented; seminal ducts are normally present and lay close to the prostatic urethra (arrows). UL, urethral lumen; Ur, urothelium; PG, prostatic glands; Ch, chorion; SM, smooth muscle; VM, verumontanum.

## 3. Results

*3.1. Histologic Features of the PU. Distance between Basal Membrane of the PU Epithelium and the Prostatic Glandular Tissue*

Six specimens with a clear depiction of the histologic characteristics of the "standard" PU were included. Median prostate volume and PUL were 45 cc (38–50) and 35 mm (32–39), respectively. Histologically, three layers of the PU were observed:

(1)   urothelium with basement membrane,
(2)   subepithelial connective tissue (SCT) (chorion),
(3)   prostatic periurethral muscular tissue (PPUM).

In the post-basal sections, the average overall thickness of the chorion is 0.29 mm, ranging between 0.3 mm and 0.28 mm for the anterior and posterior prostatic portions, respectively (Table 2a). The average distance between the chorion and prostate glands (Table 2b) is 2.5 mm (2.81 vs 2.2 mm for the anterior and posterior portions, respectively). At this level, muscle fibers are well represented (Figures 2B and 3A). These are distributed in a sleeve around the basal urethra, representing a continuation of the detrusor muscle.

**Table 2.** a. Chorion thickness at the level of the basal portion of the PU. b. Distance between basal membrane of the PU epithelium and prostate glands at the level of the basal PU.

| **a** | | | | | | | |
|---|---|---|---|---|---|---|---|
| **BASAL URETHRA Chorion Thickness** | **Anterior Right** | **Anterior Left** | **Posterior Right** | **Posterior Left** | **Average** | **Average Anterior** | **Average Posterior** |
| n.1 Specimen | 0.2 | 0.2 | 0.2 | 0.2 | 0.2 | 0.2 | 0.2 |
| n.2 Specimen | 0.3 | 0.2 | 0.2 | 0.3 | 0.25 | 0.25 | 0.25 |
| n.3 Specimen | 0.4 | 0.2 | 02 | 0.2 | 0.25 | 0.3 | 0.2 |
| n.4 Specimen | 0.3 | 0.3 | 0.2 | 0.3 | 0.27 | 0.3 | 0.25 |
| n.5 Specimen | 0.4 | 0.4 | 0.4 | 0.4 | 0.4 | 0.4 | 0.4 |
| n.6 Specimen | 0.4 | 0.4 | 0.4 | 0.4 | 0.4 | 0.4 | 0.4 |
| **b** | | | | | | | |
| **BASAL URETHRA Distance Basal Membrane PU Epithelium-Prostate Glands** | **Anterior Right** | **Anterior Left** | **Posterior Right** | **Posterior Left** | **Average** | **Average Anterior** | **Average Posterior** |
| n.1 Specimen | 2 | 2.5 | 0.2 | 2.5 | 1.8 | 2.25 | 1.35 |
| n.2 Specimen | 1.3 | 2.6 | 0.4 | 3.3 | 1.9 | 1.96 | 1.85 |
| n.3 Specimen | 4 | 4 | 0.3 | 1.7 | 2.5 | 4 | 1 |
| n.4 Specimen | 1 | 0.3 | 1 | 1 | 0.82 | 0.65 | 1 |
| n.5 Specimen | 4 | 4 | 4 | 4 | 4 | 4 | 4 |
| n.6 Specimen | 4 | 4 | 4 | 4 | 4 | 4 | 4 |

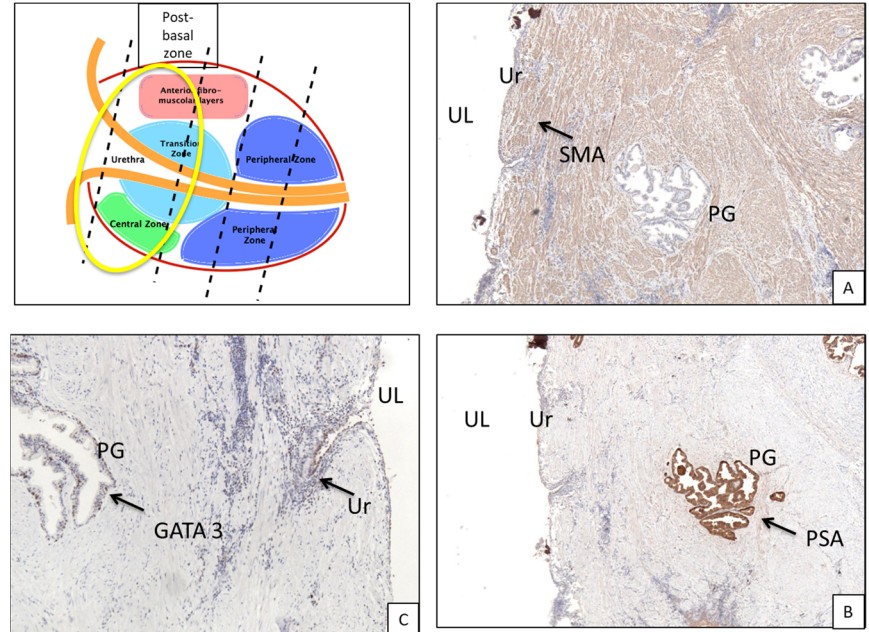

**Figure 3.** Basal portion of prostate gland from a cystoprostatectomy specimen. (**A**) Immunohisto-chemical staining for smooth muscle actin evidences the fibromuscular tissue around the prostatic urethra (arrow, 4× magnification). (**B**) Immunohistochemical staining for PSA reveals the presence of a prostatic-type gland right below the urethra epithelium (arrow, 4× magnification). (**C**) Immunohis-tochemical staining for GATA3 evidences the presence of mixed epithelium of prostatic and urothelial type (arrows, 4× magnification). UL, urethral lumen; Ur, urothelium; PG, prostatic glands; Ch, chorion; SM, smooth muscle; SMA, smooth muscle actin.

In the mid-glandular section, the average overall thickness of the chorion is 0.25 mm (Table 3a) with a mean difference of 0.04 mm between the anterior (0.27 mm) and posterior portion (0.23 mm). The prostate glands from the chorion are 1.49 mm apart (1.69 vs. 1.29 mm for the anterior and posterior portion, respectively) (Table 3b). Concerning the

PPUM, it is well represented, but decreases at the interface with the chorion, forming thin muscular fibers (Figure 4A).

**Table 3.** a. Chorion thickness at the level of the middle PU. b. Distance between basal membrane of the PU epithelium and prostate glands at the level of the middle PU.

| a | | | | | | | |
|---|---|---|---|---|---|---|---|
| **MIDDLE URETHRA Chorion Thickness** | **Anterior Right** | **Anterior Left** | **Posterior Right** | **Posterior Left** | **Average** | **Average Anterior** | **Average Posterior** |
| n.1 Specimen | 0.3 | 0.3 | 0.2 | 0.2 | 0.25 | 0.3 | 0.2 |
| n.2 Specimen | 0.3 | 0.2 | 0.2 | 0.2 | 0.22 | 0.25 | 0.2 |
| n.3 Specimen | 0.3 | 0.2 | 0.2 | 0.2 | 0.22 | 0.25 | 0.2 |
| n.4 Specimen | 0.3 | 0.3 | 0.3 | 0.3 | 0.3 | 0.3 | 0.3 |
| n.5 Specimen | 0.3 | 0,2 | 0,2 | 0.3 | 0.25 | 0.25 | 0.25 |
| n.6 Specimen | 0.3 | 0.2 | 0.3 | 0.2 | 0.25 | 0.25 | 0.25 |

| b | | | | | | | |
|---|---|---|---|---|---|---|---|
| **MIDDLE URETHRA Distance Basal Membrane PU Epithelium-Prostate Glands** | **Anterior Right** | **Anterior Left** | **Posterior Right** | **Posterior Left** | **Average** | **Average Anterior** | **Average Posterior** |
| n.1 Specimen | 1.4 | 2 | 0.4 | 1.2 | 1.25 | 1.7 | 0.8 |
| n.2 Specimen | 1.2 | 2 | 1.1 | 2 | 1.58 | 1.6 | 1.55 |
| n.3 Specimen | 4 | 2.2 | 2 | 1 | 2.3 | 3.1 | 1.5 |
| n.4 Specimen | 0.4 | 0.1 | 0.4 | 0.1 | 0.25 | 0.25 | 0.25 |
| n.5 Specimen | 0.2 | 1.5 | 2 | 3 | 1.68 | 0.85 | 2.5 |
| n.6 Specimen | 2.3 | 3 | 1.3 | 1 | 1.9 | 2.65 | 1.15 |

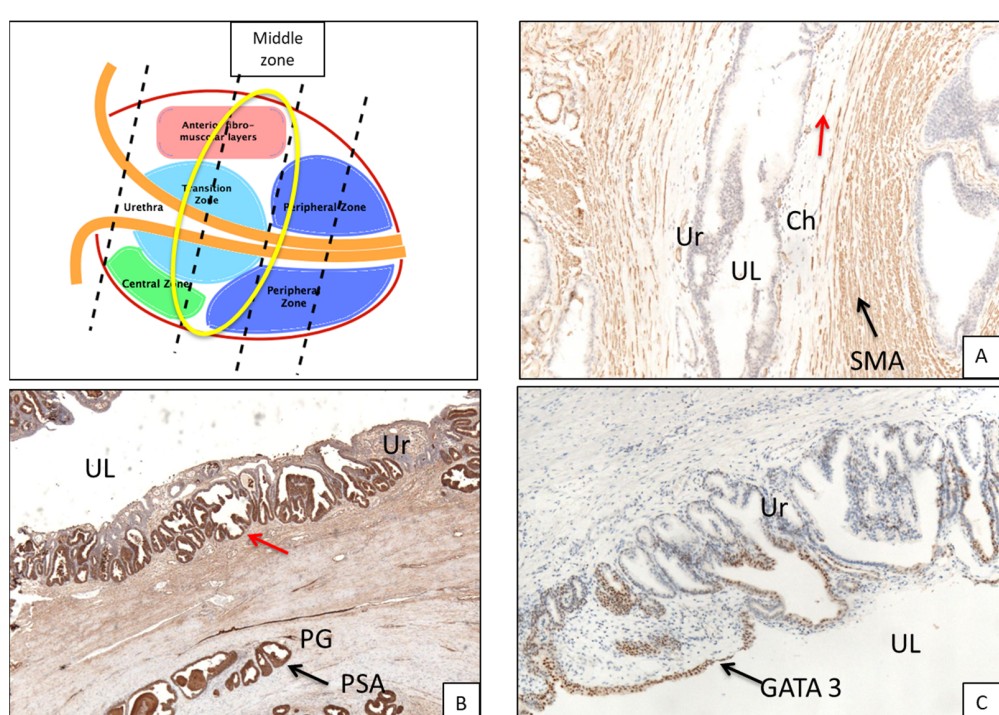

**Figure 4.** (**A**) Middle portion of urethra shows the presence of fibromuscular tissue right below the urothelium (arrow, smooth muscle actin immunohistochemistry, 4× magnification); red arrow shows small and thin muscle fibers progressively vanishing in chorion. (**B**) Immunohistochemical staining for PSA shows the presence of prostatic-type glands below the urothelial epithelium (arrow); notably, part of the urothelium is PSA-positive (red arrow, 4× magnification). (**C**) Immunohistochemical staining for GATA3 shows mixed positivity with PSA in urothelium (arrow, 4× magnification). UL, urethral lumen; Ur, urothelium; PG, prostatic glands; Ch, chorion; SM, smooth muscle; SMA, smooth muscle actin.

Finally, in the pre-apical section, the average overall thickness of the chorion is about 0.26 mm, with an average difference between the anterior (0.27 mm) and posterior portion (0.25 mm) of 0.02 mm (Table 4a). Prostate glands are encountered at an average distance of about 1.23 mm from the chorion, without a significant difference between the anterior and posterior urethral portions (1.12 mm vs. 1.34 mm, Table 4b). Concerning the PPUM, muscle fibers become more dense, and are well represented at this level with respect to the mid-glandular level. These fibers surround the urethral apex and continue to the striated portion of the external urethral sphincter (Figure 5A).

**Table 4.** a. Chorion thickness at the level of the apical PU. b. Distance between basal membrane of the PU epithelium and prostate glands at the level of the apical PU.

| a | | | | | | | |
|---|---|---|---|---|---|---|---|
| **APICAL URETHRA Chorion Thickness** | **Anterior Right** | **Anterior Left** | **Posterior Right** | **Posterior Left** | **Average** | **Average Anterior** | **Average Posterior** |
| n.1 Specimen | 0.2 | 0.2 | 0.2 | 0.2 | 0.2 | 0.2 | 0.2 |
| n.2 Specimen | 0.3 | 0.3 | 0.3 | 0.2 | 0.27 | 0.3 | 0.25 |
| n.3 Specimen | 0.3 | 0.3 | 0.3 | 0.2 | 0.27 | 0.3 | 0.25 |
| n.4 Specimen | 0.4 | 0.4 | 0.4 | 0.4 | 0.4 | 0.4 | 0.4 |
| n.5 Specimen | 0.2 | 0.2 | 0.2 | 0.2 | 0.2 | 0.2 | 0.2 |
| n.6 Specimen | 0.2 | 0.2 | 0.2 | 0.2 | 0.2 | 0.2 | 0.2 |
| b | | | | | | | |
| **APICAL URETHRA Distance Basal Membrane PU Epithelium-Prostate Glands** | **Anterior Right** | **Anterior Left** | **Posterior Right** | **Posterior Left** | **Average** | **Average Anterior** | **Average Posterior** |
| n.1 Specimen | 0.7 | 0.3 | 1 | 0.8 | 0.7 | 0.5 | 0.9 |
| n.2 Specimen | 1.7 | 1.2 | 2 | 1.5 | 1,6 | 1.45 | 1.75 |
| n.3 Specimen | 2 | 1 | 1.5 | 2 | 1.63 | 1.5 | 1.75 |
| n.4 Specimen | 0.4 | 0.4 | 0.4 | 0.4 | 0,4 | 0.4 | 0.4 |
| n.5 Specimen | 2 | 1 | 1.5 | 2 | 1.63 | 1.5 | 1.75 |
| n.6 Specimen | 1.7 | 1 | 2 | 1 | 1.43 | 1.35 | 1.5 |

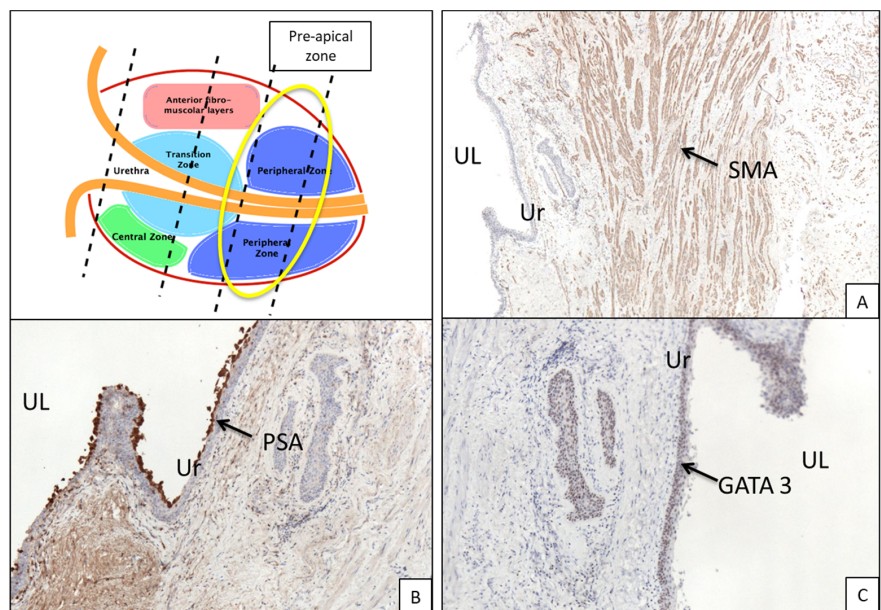

**Figure 5.** (**A**) Smooth muscle actin immunohistochemical staining shows the fibromuscular component of prostatic apex, underlining the fascicular disposition of fibers (arrow, 10× magnification). (**B**) PSA immunohistochemistry shows positivity in urethral epithelial lining (arrow, 10× magnification), and coexpression of GATA 3 (**C**) is also present (arrow, 10× magnification). UL, urethral lumen; Ur, urothelium; PG, prostatic glands; Ch, chorion; SM, smooth muscle; SMA, smooth muscle actin.

In all six specimens, the distribution of chorion thickness is homogeneous, with only slight variations, mainly at the level of the post-basal and pre-apical sections, as shown in Figure 6.

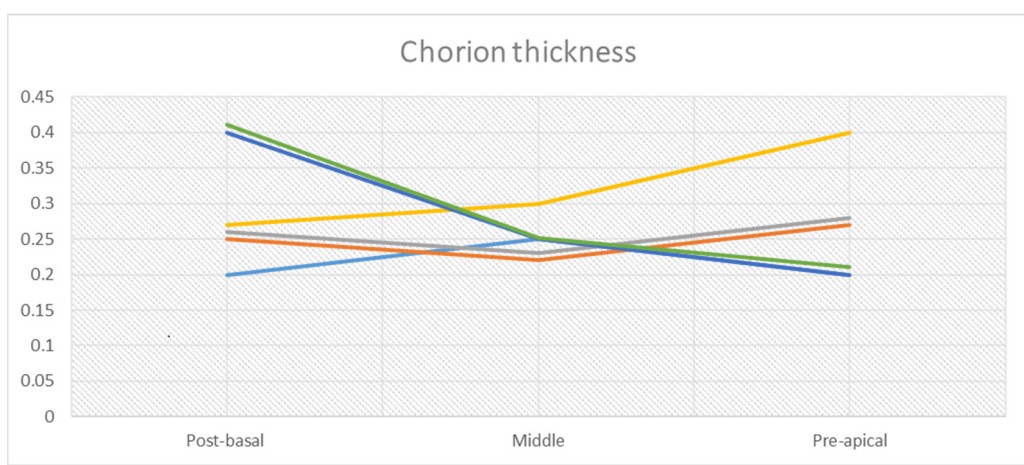

**Figure 6.** Mean chorion thickness of all six specimens, with progression from basal to apical portions.

Fibromuscular tissue is well represented in the apical and basal regions, with dense muscle fibers that represent a blending of the prostatic fibromuscular stroma with the peri-urethral muscular structures.

According to our observations, no correlation between chorion thickness and the amount (density) of periurethral fibromuscular tissue was encountered.

The overall mean distance between the basal membrane of the PU urothelium and the prostate glands is 1.74 mm (Figure 7). A more variable distribution of this distance was observed at the level of the basal sections; the differences become less remarkable at the level of the mid-prostate and in the apical section of the gland. In all specimens, this distance lessens proceeding from the base of the prostate to the middle urethral and apical portions of PU (2.5 vs. 1.49 vs. 1.23 mm, respectively).

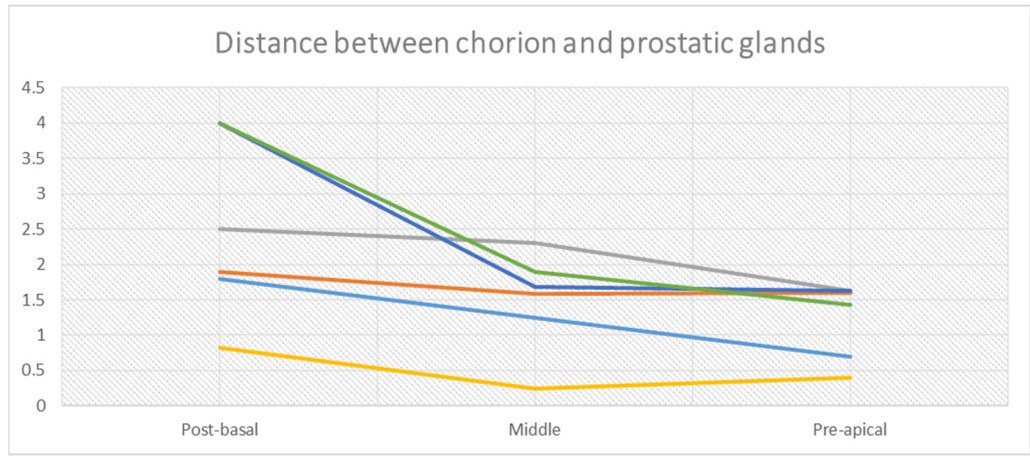

**Figure 7.** Mean distance between prostatic urothelium and prostatic glands of all six specimens (progressively from basal to apical portions).

*3.2. Immunohistochemistry (Figures 3B,C and 5B,C)*

Immunohistochemical analysis of PSA and GATA3 showed coexpression in both the urethral epithelium and periurethral glands of all sections (basal, PSA 70% and GATA-3 30%; middle, 70% PSA and 30% GATA3; apical 40% PSA and 60% GATA3).

## 4. Discussion

Long-term continence rates after RALP range between 82.1% and 97% [8,9], depending on the definition of incontinence and the methodology used to assess its magnitude [10]. However, high rates of early postoperative urinary incontinence (UI) (i.e., in the first 2–6 months after surgery) persist [11–13], causing significant bother/distress among men undergoing RALP [14].

Several techniques have been developed in order to enhance continence recovery after RALP that can be summarized in three major categories: preservation, reconstruction, and reinforcement [15]. The aim of these techniques is to maintain the intactness, or reply the functionality, of the periprostatic anatomic structures [16].

In addition to bladder neck preservation [4,17], several studies support an association between the length of membranous urethra preserved and continence recovery [18–21]. A similar benefit for PUP was suggested by a retrospective series of studies in which a portion of the apical PU was spared during open RP [22,23] or RALP [5].

Concerning EPUP, the technique and outcomes were presented in a previous study by Nunez Bragayrac LA et al. [6]. As described by the authors, with the prostate in cephalad retraction, a cranially-oriented circumferential blunt dissection, with occasional cold scissor division, delineates the plane between the outer urethra striated muscle and prostatic apex. Further intraprostatic dissection exposes the inner longitudinal muscle fibers that are deeper than the outer striated muscle fibers, which terminate in the prostate apex. The dissection is continued further intraprostatically; a deeper intramuscular plane within the longitudinal fibers can be followed to ensure a negative margin with prostatic tissue. This approach "telescopes" the urethra muscle, as a longer segment is preserved.

All EPUP patients underwent a preoperative multiparameter prostate MRI that showed no tumor involvement of the apical PU.

Using this technique, urethral sparing was achieved till the prostate mid-gland or base. In two patients, the transected bladder neck was reached, indicating a complete preservation of the entire PU. The amount of spared PUL achieved by this technique ranged between 2.5 and 6 cm, with a median value of 4 cm. In a multivariable analysis, the EPUP approach was independently associated with earlier continence recovery; a longer PUL provided a higher functional benefit in terms of continence recovery.

Although performed by the same surgeon, a complete EPUP was achieved only in 2 of 48 patients. At the completion of the vesicourethral anastomosis, the surgeon added stitches that tended to shorten or collapse the preserved urethra in order to avoid tearing; thus, no knowledge was provided either on eventual kinking of the preserved urethral stump or on the vascularization of the preserved segment. Finally, nearly 8% of EPUP patients had a detectable PSA (>0.1 ng/dL) at a median follow-up of 12 months, raising concerns about the possible permanence of PSA-producing tissue. The same authors suggest that the oncologic safety of EPUP requires further investigation, and also underline the absence of validation of the preoperative mpMRI for PU neoplastic involvement.

In our study, we evaluated the morphology of the PU. It can be summarized with the presence of a pseudo-stratified urothelial-type epithelium, sometimes mixed with epithelium with prostatic differentiation. The urethral epithelium of the prostate does not have a regular course; it may present pseudopapillary protrusions, areas of thickening, and glandular ramifications that combine to create a non-linear "tube". Others have also described that the proximal urethral segment forms an angulation of about 35 degrees anterior to the course of the distal urethral segment [24].

Below the basal lamina of the PU epithelium, there is a thin lamina of loose connective tissue that is interposed between the epithelium itself and the fibromuscular stroma of the prostate (that itself hosts the prostate glands); this space can be defined as the chorion or sub-epithelial tissue. The fibromuscular stroma does not arrange itself continuously around the chorion in a precise manner, maintaining a clear interface, but sometimes tends to "fray" within the sub-epithelial tissue, creating an ill-defined passage point.

The thickness of the aforementioned planes measures fractions of millimeters.

In the evaluated specimens, a rather homogeneous distribution of the thickness of the chorion was observed between the base and apex of the prostate, ranging between 0.2 and 0.4 mm. The distance between the urothelium of the PU and prostatic glandular tissue showed a more variable distribution of average values (Figure 3), with the greatest variability in the basal portions and a progressively decreasing gradient of variability towards the apical ones. At the level of the post-basal sections, the distance between the PU epithelium and the prostate glands ranged between 0.82 and 4mm, while at the pre-apical sections the distances between the PU epithelium and glandular tissue lay within a narrower range, between 0.4 and 1.63 mm.

In conclusion, the wide variability and narrowness of the distance between the PU and the glandular stroma, together with the irregular distribution of the PUM, probably do not allow for effective exclusion of prostate glandular tissue from the peri-urethral resection margin.

Furthermore, the results of the immunohistochemistry confirm the mixed phenotype of urethral epithelial cells, most evident in the mid-apical sections. These PSA-producing cells may be partially responsible for the cases of PSA persistence described in [6]. Similar to our results, Son DY et al. [25] observed three types of PU glands: urethral mucosal, prostatic acinar, and mixed. The proximal segment of the PU and the bladder neck consisted mostly of the urethral mucosal type, whereas the distal segment and apical margins consisted mostly of the prostatic acinar type. PSA was expressed in secretory cells in prostatic acinar and mixed types.

These results indicate that PSA-expressing cells are abundant in all segments of the PU till the apical margin, and may be responsible for postoperative PSA persistence [26].

The scarce rate of complete PU preservation described in [6] may also be due to individual anatomical variations, both in the PU and periurethral glandular density, in patients affected by benign prostatic hyperplasia [27]; these variations may further hinder the standardization of EPUP technique.

Another aspect of PU preservation to consider is its vascularization. The PU is supplied by urethral branches of the prostatic artery, which wrap the urethral tube in its basal, middle, and apical portions, both anteriorly and posteriorly [28]. Thus, it seems not possible to preserve the vascular supply of the preserved PU, while the preservation of a long urethral tube may be prone to kinking during micturition. Immunohistochemical studies with a vascular wall antibody (CD31) may clarify this issue.

The main limitation of this histologic/morphologic study was the limited sample. However, the included specimens demonstrate a homogeneous distribution of the thickness of the chorion and distance between the chorion and prostatic glands; consequently, the analysis of more specimens seems superfluous.

## 5. Conclusions

The exiguity of thickness of the PU chorion, the short distance from glandular tissue, and the coexistence of PSA-expressing cells in the PU epithelium raise important concerns about the oncologic safety of EPUP, since, during the surgical maneuvers, periurethral prostatic glands may be left in place and may be the basis of PSA-persistence after surgery. The results of our histologic study should be further validated by adequately designed clinical studies evaluating the oncologic safety of EPUP in clinical practice.

**Author Contributions:** Conceptualization, A.D.A.; methodology, F.A. and F.B.; software, G.C. and F.B.; validation, A.D.A., R.G. and T.P.; formal analysis, U.A.; investigation, G.C. and A.M.; data curation, G.C. and F.B.; writing—original draft, A.D.A. and F.B. All authors made a significant contribution to the findings and methods in the paper. This work has not already been published and has not been submitted simultaneously to any other journal. All authors have read and agreed to the published version of the manuscript.

**Funding:** This research received no external funding.

**Institutional Review Board Statement:** The study was conducted in accordance with the Good Clinical Practice rules and with the ethical principles contained in the Declaration of Helsinki, as revised in 2013. All patients provided written informed consent with guarantees of confidentiality. Pathologic data were prospectively collected and inserted in a customized, IRB-approved database (Fondazione PTV Policlinico Tor Vergata, protocol code 203.22). Data were analyzed anonymously after removal of patient identifiers.

**Informed Consent Statement:** Informed consent was obtained from all subjects involved in the study.

**Data Availability Statement:** Data is unavailable due to privacy or ethical restrictions.

**Conflicts of Interest:** The authors declare no conflict of interest.

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
