# Peer review of "“Urethral-Sparing” Robotic Radical Prostatectomy: Critical Appraisal of the Safety of the Technique Based on the Histologic Characteristics of the Prostatic Urethra"

_curroncol, doi:10.3390/curroncol30010082_

Round 1

Reviewer 1 Report

This study aims to provide a histologic description of the prostatic urethra in order to evaluate the safety of its complete preservation during RALP. I think that the aim has to be better explained. Nevertheless, the study is of interest and a few changes have to be done before publications.

Please specify the characteristics of the study, the ethical committee approval, and the selection of patients.

Please check the language. E.g.: Line 220 please complete the sentence: the technique and outcomes have been presented in… (…a previous study…; …By Nunez et al….). Moreover, use always the same abbreviations (RALP/RARP)

Author Response

This study aims to provide a histologic description of the prostatic urethra in order to evaluate the safety of its complete preservation during RALP. I think that the aim has to be better explained. Nevertheless, the study is of interest and a few changes have to be done before publications.

We thank the reviewer for his/her comments.

Please specify the characteristics of the study, the ethical committee approval, and the selection of patients.

We apologize to the reviewer, but probably during the uploading process a prefinal version of the manuscript has been uploaded. We herein attach the ethics for the study as in the manuscript lines 179-184:

Ethics

The study was conducted in accordance with the Good Clinical Practice rules and with the ethical principles contained in the Declaration of Helsinki as revised in 2013. All patients provided written informed consent with guarantees of confidentiality. Pathologic data was prospectively collected and inserted in a customized, IRB-approved database (IRB approval PTV-203.22). Data was analyzed anonymously after removal of patient identifiers.

Concerning the characteristics of the study and the selection of the patients, we only selected prostate specimens obtained by routine surgery and our goal was to study the histologic characteristics of the “standard” prostatic urethra, excluding cases of anatomic and histologic variations, caused by both pathological conditions and individual anatomical changes that could modify the interpretation of the histologic results. The six cases were selected from a pool of 24 prostatic specimens since they offered a “reliable” interpretation of the histologic characteristics of the PU.

These issues have been clarified in the materials and methods section in lines 144-149 as follows:

Twenty-four prostatic specimens of RP and cystoprostatectomy, performed for the treatment of prostate and muscle-invasive bladder cancer, respectively, were obtained from the routine urological clinical practice and were histologically analyzed. Eighteen of them were characterized by PU distortion by hyperplastic nodules or with PU neoplastic involvement and were excluded fromn the final analysis. Consequently, six specimens with a “reliable” depiction of the histologic characteristics of the “standard” PU were included.

Please check the language. E.g.: Line 220 please complete the sentence: the technique and outcomes have been presented in… (…a previous study…; …By Nunez et al….). Moreover, use always the same abbreviations (RALP/RARP)

We thank the reviewer, we corrected the sentence as suggested and we homogenized the use of RALP throughout the manuscript.

Reviewer 2 Report

It is a study investigating the histologic description of the prostatic urethra after radical prostatectomy and radical cystoprostatectomy.

Introduction: The background of the role of prostatic urethra preservation in the continence after the radical prostatectomy is well described. The aim is clearly stated.

Material and methods: Prostatic Urethra specimens after both radical prostatectomy and cystoprostatectomy were evaluated. The histological examination was adequately detailed.

The preservation of prostatic urethra after cystoprostatectomy depends on the urine diversion, consequently, the inclusion of cystoprostatectomy may be considered a drawback.

Neither oncological nor functional data were recorded. How did you evaluate the oncological safety?

Results: The exact number of examined prostatic urethras (six) should also be stated in the results section, not only in the abstract. The sample size is too small. There is no correlation with the functional or oncological outcomes.

Discussion: The main points were adequately described. The limitations of the study were not mentioned.

Conclusions: How did you prove the oncological risks?

The figures and the tables are very comprehensive.

Author Response

It is a study investigating the histologic description of the prostatic urethra after radical prostatectomy and radical cystoprostatectomy.

Introduction: The background of the role of prostatic urethra preservation in the continence after the radical prostatectomy is well described. The aim is clearly stated.

We thank the reviewer for his/her comments.

Material and methods: Prostatic Urethra specimens after both radical prostatectomy and cystoprostatectomy were evaluated. The histological examination was adequately detailed.

We thank the reviewer for his/her comments.

The preservation of prostatic urethra after cystoprostatectomy depends on the urine diversion, consequently, the inclusion of cystoprostatectomy may be considered a drawback.

Aim of the study was to evaluate the histology of the intraprostatic urethra, that is completely preserved in cystectomies independently of the degree of preservation of the membranous urethra (that may depend itself on the urinary diversion). Furthermore, the cystectomy specimens allowed for a clear description of the basal and post-basal prostatic urethra, since the electrocautery artifacts that are caused during the bladder neck dissection in RP are avoided.

Neither oncological nor functional data were recorded. How did you evaluate the oncological safety?

Aim of the study was to provide a histologic (combined to immunohistochemical evaluation) of the PU. The study, after having performed several measurements, suggests that the chorion surrounding the PU is exile and the distance from the chorion itself to the prostatic glands is very short. Moreover, a coexistence of PSA-expressing cells in the PU epithelium has been encountered. These elements raise important concerns about the oncologic safety of EPUP, since during the surgical maneuvers periurethral prostatic glands may be left in place and may be at the basis of the PSA-persistence after surgery. These issues have been introduced in the conclusion section.

Results: The exact number of examined prostatic urethras (six) should also be stated in the results section, not only in the abstract.

Performed.

The sample size is too small. There is no correlation with the functional or oncological outcomes.

We agree that the sample size is small, however the six specimens were selected from twenty-four prostatic specimens of RP and cystoprostatectomy, performed for the treatment of prostate and muscle-invasive bladder cancer, respectively. Eighteen of them were characterized by PU distortion by hyperplastic nodules or with PU neoplastic involvement and were excluded from the final analysis. Consequently, six specimens with a “reliable” depiction of the histologic characteristics of the “standard” PU were included.

Furthermore, our study does not prove oncologic risks, it only raises concerns based on histology about the oncologic safety of EPUP. The results of our histologic study should be further validated by adequately designed clinical studies that should evaluate the oncologic safety of the EPUP in the clinical practice.

Discussion: The main points were adequately described. The limitations of the study were not mentioned.

The main limitation of this histologic/morphologic study was the limited sample. However, the specimens demonstrate a homogeneous  distribution of the thickness of the chorion and the distance between chorion and prostatic glands and consequently, the analysis of more specimens may be superfluous. Please see lines 320-323.

Conclusions: How did you prove the oncological risks?

As stated before, aim of the study was to provide a histologic (combined to immunohistochemical evaluation) of the PU. The study, after having performed several measurements, suggests that the chorion surrounding the PU is exile and the distance from the chorion itself to the prostatic glands is very short. Moreover, a coexistence of PSA-expressing cells in the PU epithelium has been encountered. These elements raise important concerns about the oncologic safety of EPUP, since during the surgical maneuvers periurethral prostatic glands may be left in place and may be at the basis of the PSA-persistence after surgery.

Thus our study does not prove oncologic risks, it only raises concerns based on histology about the oncologic safety of EPUP. The results of our histologic study should be further validated by adequately designed clinical studies that should evaluate the oncologic safety of the EPUP in the clinical practice.

The figures and the tables are very comprehensive.

Round 2

Reviewer 2 Report

All the required revisions were performed. I have no additional comments. The study should be considered for publication.